# Molecular and Metabolic Mechanism of Low-Intensity Pulsed Ultrasound Improving Muscle Atrophy in Hindlimb Unloading Rats

**DOI:** 10.3390/ijms222212112

**Published:** 2021-11-09

**Authors:** Lijun Sun, Shasha An, Zhihao Zhang, Yaling Zhou, Yanan Yu, Zhanke Ma, Xiushan Fan, Liang Tang, Jianzhong Guo

**Affiliations:** 1Institute of Sports Biology, Shaanxi Normal University, Xi’an 710119, China; sunlijun@snnu.edu.cn (L.S.); as@snnu.edu.cn (S.A.); zhangzhihao@bsu.edu.cn (Z.Z.); zhouyl184@snnu.edu.cn (Y.Z.); limnan128@snnu.edu.cn (Y.Y.); mazhanke8888@163.com (Z.M.); xshfan@snnu.edu.cn (X.F.); 2Shaanxi Key Laboratory of Ultrasonics, Shaanxi Normal University, Xi’an 710119, China

**Keywords:** muscle atrophy, low intensity pulsed ultrasound, myostatin, metabonomics

## Abstract

Low-intensity pulsed ultrasound (LIPUS) has been proved to promote the proliferation of myoblast C2C12. However, whether LIPUS can effectively prevent muscle atrophy has not been clarified, and if so, what is the possible mechanism. The aim of this study is to evaluate the effects of LIPUS on muscle atrophy in hindlimb unloading rats, and explore the mechanisms. The rats were randomly divided into four groups: normal control group (NC), hindlimb unloading group (UL), hindlimb unloading plus 30 mW/cm^2^ LIPUS irradiation group (UL + 30 mW/cm^2^), hindlimb unloading plus 80 mW/cm^2^ LIPUS irradiation group (UL + 80 mW/cm^2^). The tails of rats in hindlimb unloading group were suspended for 28 days. The rats in the LIPUS treated group were simultaneously irradiated with LIPUS on gastrocnemius muscle in both lower legs at the sound intensity of 30 mW/cm^2^ or 80 mW/cm^2^ for 20 min/d for 28 days. C2C12 cells were exposed to LIPUS at 30 or 80 mW/cm^2^ for 5 days. The results showed that LIPUS significantly promoted the proliferation and differentiation of myoblast C2C12, and prevented the decrease of cross-sectional area of muscle fiber and gastrocnemius mass in hindlimb unloading rats. LIPUS also significantly down regulated the expression of MSTN and its receptors ActRIIB, and up-regulated the expression of Akt and mTOR in gastrocnemius muscle of hindlimb unloading rats. In addition, three metabolic pathways (phenylalanine, tyrosine and tryptophan biosynthesis; alanine, aspartate and glutamate metabolism; glycine, serine and threonine metabolism) were selected as important metabolic pathways for hindlimb unloading effect. However, LIPUS promoted the stability of alanine, aspartate and glutamate metabolism pathway. These results suggest that the key mechanism of LIPUS in preventing muscle atrophy induced by hindlimb unloading may be related to promoting protein synthesis through MSTN/Akt/mTOR signaling pathway and stabilizing alanine, aspartate and glutamate metabolism.

## 1. Introduction

In the long-term lack of mechanical stimulation, the size, strength, volume and cross-sectional area of skeletal muscle will be significantly reduced. Due to the influence of microgravity environment, the weight of limb skeletal muscle of astronauts in space flight for a long time will be reduced [1,2], among which the loss proportion of lower limb skeletal muscle is greater, and the mass loss speed of gastrocnemius muscle is faster [1]. In addition, muscle strength will be significantly decreased [3,4,5,6,7]. Muscle atrophy can also be observed in long-term bedridden patients. For example, paralysis patients will have 6–40% muscle mass reduction [8], resulting in muscle strength unable to recover, affecting activities of daily living [9]. It is projected that in 2050, 20% of the world’s population over 60 will suffer from sarcopenia and by 2150, this percentage will increase to 33% of the population [10]. Muscular atrophy is associated with metabolic abnormalities, including changes in insulin sensitivity, increased fat and connective tissue infiltration in skeletal muscle, decreased hormone levels, decreased mitochondrial activity, resulting in impaired oxidative defense, and severe bone loss resulting in osteoporosis [11,12].

Low intensity pulsed ultrasound (LIPUS) is a kind of low-intensity physical therapy, which has been widely proved to have a positive effect on bone healing. Because of its noninvasive and targeted characteristics, it is gradually applied, especially in clinical auxiliary stimulation of tissue regeneration [13]. Our previous studies have reported that LIPUS significantly promoted exercise-induced muscle hypertrophy and prevented muscle atrophy caused by type 1 diabetes [14]. In addition, LIPUS was reported to active the proliferation of myoblasts [15]. In addition, under different frequency and intensity of low intensity pulsed ultrasound stimulation, it can be concluded that 3 MHz and 1 W/cm^2^ stimulation can promote cell proliferation to the maximum extent. At 500 mW/cm^2^, 1 MHz stimulation maximized differentiation and subsequent multinucleated myotubes [16]. However, the issue about whether LIPUS can reduce muscle atrophy caused by tail suspension and its mechanisms was not clear [14,17].

Metabonomics is a new method to describe small molecules in organisms, cells and tissues. Since metabolites represent the downstream expression of genome, transcriptome and proteome, it is most beneficial to study them for disease risk and phenotypic changes and endogenous variation [18]. In the aging model of muscle atrophy, the type of fast muscle fiber involved in glycolysis will decrease rapidly [19]. A shift in fuel metabolism from lipid fuels to glucose has been found in models of space flight [20], hind limb suspension [21], bed rest and aging [22,23]. In addition, the accumulation of triglycerides in atrophic muscles was also found [24,25,26]. Phospholipids in skeletal muscle also change during atrophy [27]. Some studies have reported the changes of metabolites, such as amino acids, intermediate products of TCA cycle and oxidative stress markers in skeletal muscle [28,29]. However, the effect of LIPUS on the metabonomics of muscle atrophy has not been reported.

In this study, we investigate the effects of LIPUS on muscle atrophy in tail suspended rats, and analyze the underlying mechanism from the perspective of metabonomics. Besides, myostatin (MSTN), a key negative regulator of muscle growth and development, which has strong catabolic and anti-anabolic effects on skeletal muscle were also evaluated.

## 2. Result

### 2.1. Body Weight and Muscle Weight

Body weight and gastrocnemius weight were measured at the end of the experiment (Figure 1). Compared with NC group, both of the body weight and gastrocnemius weight in UL group, UL + 30 mW/cm^2^ and UL + 80 mW/cm^2^ groups were significantly decreased (*p* < 0.01; *p* < 0.01; *p* < 0.05). Compared with UL group, gastrocnemius weight of group UL + 80 mW/cm^2^ increased significantly (*p* < 0.05).

### 2.2. Mean Cross Sectional Area of Muscle Fiber

The average cross-sectional area of fast muscle fiber was shown in Figure 2A. Compared with UL group, UL + 80 mW/cm^2^ group showed a significant increase (*p* < 0.05). As shown in Figure 2B, compared with the UL group, the average cross-sectional area of slow muscle fiber in UL + 30 mW/cm^2^ and UL + 80 mW/cm^2^ group was significantly higher (*p* < 0.01, *p* < 0.01). The average cross-sectional area of muscle fiber was shown in Figure 2C. Compared with NC group, UL group showed a significant decrease (*p* < 0.01). Both 30 mW/cm^2^ and 80 mW/cm^2^ LIPUS significantly increased the average cross-sectional area of muscle fiber (*p* < 0.01, *p* < 0.01) compared with UL group.

### 2.3. Muscle Tension

Muscle tension in vivo was detected (Figure 3). The results showed that all of the muscle tension in the UL, UL + 80 mW/cm^2^ and UL + 30 mW/cm^2^ group decreased significantly under different frequencies of electric stimulation. Compared with UL group, the muscle tension of UL + 80 mW/cm^2^ and UL + 30 mW/cm^2^ groups increased, and the effect of UL + 80 mW/cm^2^ was slightly better than UL + 30 mW/cm^2^, but there was still no significant difference.

### 2.4. Cell Proliferation and Differentiation

As shown in Figure 4A, the number of nuclei of C2C12 cells in the 30 mW/cm^2^ LIPUS group was apparently more than that in the other two groups. Figure 4C showed that 30 mW/cm^2^ LIPUS significantly increased the number of C2C12 cells (*p* < 0.01), and the effect is higher than 80 mW/cm^2^ (*p* < 0.05) (Figure 4C). As shown in Figure 4B, compared with the NC group, 30 mW/cm^2^ and 80 mW/cm^2^ LIPUS significantly promoted cell proliferation (*p* < 0.01; *p* < 0.05), and the effect of 30 mW/cm^2^ is better than that of 80 mW/cm^2^ (*p* < 0.01).

HE staining images (Figure 5A) showed that myotube formation was observed in all groups, which was more obvious in LIPUS-30 mW/cm^2^ treatment group. Quantitative analysis (Figure 5A(a)) also showed that the number of myotubes in LIPUS-30 mW/cm^2^ group was significantly greater than NC and LIPUS-80 mW/cm^2^ groups (*p* < 0.01). From Figure 5B, the microfilaments in LIPUS group were obviously arranged in parallel and linear, which was more conducive to the differentiation of myoblasts and the formation of myotubes. However, there was no significant difference in length/width ratio (Figure 5B(b)) among the three groups. In addition, the protein expression of myotube marker MyHC3 was also detected. As shown in Figure 5C, LIPUS significantly increased the expression of MyHC3 of C2C12 cells, and the promotion effect of 30 mW/cm^2^ is better than 80 mW/cm^2^ (*p* < 0.01). The original images for Western blots were shown in Appendix A.

### 2.5. The Expressions of AKT, mTOR and MSTN and Its ReceptorActrIIB

The mRNA and protein expression of MSTN in each group is shown in Figure 6A,C. Compared with NC group, the mRNA and protein expression of MSTN in UL group was significantly increased (*p* < 0.01, *p* < 0.01). However, LIPUS at 30 mW/cm^2^ and 80 mW/cm^2^ significantly decreased the mRNA and protein expression of MSTN (*p* < 0.01, *p* < 0.01; *p*< 0.01, *p* < 0.01). In addition, the MSTN mRNA expression in UL+80 mW/cm^2^group was significantly lower than that in UL + 30mW/cm^2^ group (*p* < 0.05). The mRNA level of ActrIIB is shown in Figure 6B and there was no significant difference among the groups. The protein expression of ActrIIB is shown in Figure 6D. The protein expression of ActrIIB in UL group was significantly higher than that in NC group (*p* < 0.01). In addition, 80 mW/cm^2^ LIPUS significantly down regulated its expression (*p* < 0.05). The protein expression level of Akt is shown in Figure 6E. Compared with NC group, UL group significantly decreased the protein expression of Akt (*p* < 0.01). Compared with UL group, UL+30 mW/cm^2^ group and UL+80 mW/cm^2^ group significantly increased the expression level of Akt (*p* < 0.01, *p* < 0.01)). Figure 6F shows the protein expression level of mTOR. Compared with NC group, the protein expression level of mTOR in UL group was decreased significantly (*p* < 0.01). Compared with UL group, UL+80 mW/cm^2^ group significantly up regulated the expression level of mTOR (*p* < 0.01). The meltcurve and amplification curve of RT-PCR and and original images for Western blots were shown in Appendix A.

### 2.6. GC-MS Analysis of Metabolic Profiling

The stability and repeatability of GC-MS system for large-scale sample analysis were confirmed by mixed quality control sample analysis (QC) and internal standard retention time (RT). In the whole analysis process, four quality control samples and internal standard serum were analyzed. Principal component analysis (PCA) score chart, including all test and quality control samples, showed that the characteristics of quality control samples were closely clustered. Therefore, the stability and repeatability of the method are considered acceptable. Typical total ion chromatograms (TIC) of serum samples from NC group, UL group, UL + 30 mW/cm^2^ group and UL + 80 mW/cm^2^ group are shown in Figure 7A–C. Subsequently, PCA was performed to obtain a comprehensive view of metabonomics, and unsupervised multivariate data analysis was performed to visualize trends and outliers in NC group, UL group, UL + 30 mW/cm^2^ group and UL + 80 mW/cm^2^ group. As shown in Figure 7D–F, the score chart is not clearly separated. The permutation test is used to verify the model. R^2^Y and Q^2^ of the OPLS-DA model were 0.994 and -0.0385 (Figure 7G–J), respectively, indicating that the model was both reliable and predictive. In order to identify the variables leading to this huge separation, the importance of variables in the projection (VIP) statistics of OPLS-DA model and t test between the two groups (*p* < 0.05) was used to pre select variables. KEGG and MS library were used to analyze the metabolites by GC-MS. The metabolites must meet the conditions of VIP > 1 and *p* < 0.05. A total of 12 metabolites were identified in NC group and UL model group, 10 metabolites were produced in UL + 30 mW/cm^2^ group and UL model group, and 13 metabolites were produced in UL + 80 mW/cm^2^ group and UL model group (Table 1). Then, the path impact value is calculated through the path topology analysis with the thresholds of 0.2. Three metabolic pathways (phenylalanine, tyrosine and tryptophan biosynthesis; alanine, aspartate and glutamate metabolism; glycine, serine and threonine metabolism) were selected as important metabolic pathways for UL effect (Figure 8A), while one metabolic pathway (alanine, aspartate and glutamate metabolism) was selected as important metabolic pathway for UL + 30 mW/cm^2^ effect (Figure 9A). As shown in Figure 10A, two metabolic networks (glycine, serine and threonine metabolism; alanine, aspartate and glutamate metabolism) changed after UL + 80 mW/cm^2^.

As shown in Figure 8A(a–c), the potential biomarkers of three metabolic networks (phenylalanine, tyrosine and tryptophan biosynthesis; alanine, aspartate and glutamate metabolism; glycine, serine and threonine metabolism) significantly decreased after hindlimb unloading (L-thyrosine; L-alanine; Succinate; L-aspartate; Citric acid; L-serine) (*p* < 0.05; *p* < 0.05; *p* < 0.05; *p* < 0.05; *p* < 0.05; *p* < 0.01). The 30 mW/cm^2^ LIPUS may improve metabolic diseases caused by alanine, aspartate and glutamate metabolism disorder by increasing L-aspartate and Myo-inositol (*p* < 0.01; *p* < 0.01) (Figure 9A,A(b)). Similarly, 80 mW/cm^2^ LIPUS affected the metabolic level of hind limb unloading by improving L-aspartic acid; cholesterol; glyceric acid; xylitol and myo-inositol in glycine, serine and threonine metabolism and alanine, aspartate and glutamate metabolism (*p* < 0.05; *p* < 0.01; *p* < 0.05; *p* < 0.01; *p* < 0.01) (Figure 10A(b,c)). It is suggested that LIPUS can restore the imbalance metabolism of UL rats (Table 1).

## 3. Discussion

Long term lack of gravity can lead to skeletal muscle atrophy [30]. LIPUS can promote the differentiation of myoblasts and promote the regeneration of muscle fibers [15,16,31]. Therefore, we investigated whether LIPUS could prevent skeletal muscle atrophy induced by hindlimb unloading. Our results show that LIPUS can effectively reduce muscle atrophy caused by unloading of hind limbs, and then improve skeletal muscle function. These mechanisms may be related to MSTN/Akt/mTOR protein synthesis pathway. In addition, LIPUS can improve the three metabolic pathways (phenylalanine, tyrosine and tryptophan biosynthesis; alanine, aspartate and glutamate metabolism; glycine, serine and threonine metabolism), which were selected as important metabolic pathways for hindlimb unloading effect of atrophic rats. However, 30 mW/cm^2^ LIPUS improved one metabolic pathway (alanine, aspartate and glutamate metabolism) and 80 mW/cm^2^ LIPUS improved two metabolic pathways (alanine, aspartate and glutamate metabolism; glycine, serine and threonine metabolism).

After unloading the hind limbs, the body weight and muscle weight decreased by 25–55%, among which the lower limb skeletal muscle lost a larger proportion, and the gastrocnemius muscle was lost faster [32,33]. In the 115–197-day space flight, astronauts lost 17% of gastrocnemius weight, 17% of soleus weight and 10% of quadriceps weight [34]. Toshinori et al. [35] also found similar results in rats with hindlimb unloading. Our results showed that hindlimb unloading resulted in significant weight loss and wet weight loss of gastrocnemius muscle. However, 80 mW/cm^2^ LIPUS significantly prevented the weight loss of gastrocnemius muscle. In addition to the significant decrease of body weight and muscle mass, muscle fiber cross-sectional area and fiber type also changed [36,37]. Fiber types have different sensitivities to different pathophysiological attacks. For example, in a model of amyotrophic disease, type II fibers are more likely to be consumed than type I fibers. Or it will change from type I and type IIA to type IIB and type IIx [35]. In this study, the average muscle fiber area had significant atrophy. However, LIPUS at both 30 mW/cm^2^ and 80 mW/cm^2^ improved this situation in varying degrees. In terms of function, a large amount of muscle strength will be significantly lost after muscle discontinuation [38]. For example, after 17 days of space flight, a 10% decrease in skeletal muscle strength can be observed [5]. In our study, the muscle strength of rats decreased significantly after unloading hindlimb. However, LIPUS had a tendency to prevent the muscle strength loss, but there was no significant difference. The effect of LIPUS on myoblast C2C12 was also evaluated. The results showed that LIPUS could significantly promote the proliferation of C2C12, which was consistent with the results of Imashiro et al. [39]. Similar to our experiment, after stimulating C2C12 for 5 min at 3.6 MHz center frequency, 100 Hz repetition rate and 27.8% duty cycle, LIPUS significantly promoted cell proliferation [40]. Salgarella et al. [16] found that 1 MHz stimulation maximized cell differentiation and subsequent formation of polynuclear myotubes at an intensity of 500 mW/cm^2^. In addition, Viviane et al. [41]. reported that the ultrasound treated cells had thicker myotubes. Consistent with these previous conclusions, our results also showed that LIPUS effectively promoted the differentiation of myoblasts and the formation of myotubes, of which 30 mW/cm^2^ LIPUS had better effect than 80 mW/cm^2^.

MSTN plays a key role in limiting skeletal muscle growth. Its targeted or naturally occurring mutations have been shown to cause muscle growth in cattle, rabbits, rats and humans in a “double” phenotype [42,43,44,45,46]. High expression of MSTN is usually found in muscle atrophy or senile muscle diseases [47,48,49]. It has been suggested that MSTN can control muscle quality by inhibiting cell proliferation and DNA and protein synthesis of mouse myoblast C2C12 [50]. Under the influence of MSTN receptor, muscle atrophy may occur [51]. Marzucanassr et al. [52] found an increase in MSTN expression in the muscle of rats with hindlimb unloading. In our previous hindlimb unloading rat model, LIPUS effectively inhibited the expression of MSTN in quadriceps femoris muscle of rats with muscular atrophy [53]. Similarly, LIPUS significantly inhibited MSTN expression in type 1 diabetic rats, exercise-induced muscle hypertrophy rats and ovariectomized rats [14,17,54]. In this study, LIPUS also significantly inhibited the expression of MSTN and its receptor at mRNA and protein levels, which may be related to the mechanism of LIPUS improving muscle atrophy.

Akt is the metabolic control point of many muscle mass change diseases, which not only promote the formation of new proteins, but also inhibit the hydrolysis of proteins [55]. The absence of Akt leads to the decrease of muscle volume and affects the transformation of muscle fiber type in mice [56,57]. Significant muscle atrophy was found in Akt subtype knockout mice [58]. Lai et al. [59] demonstrated that the expression of inducible active Akt increased the weight of quadriceps femoris in mice by 73% within 14 days. In L6 myoblasts, Akt subtypes were found to promote the transport of glucose and amino acids to cells and promote protein synthesis [60]. In our experiment, the hindlimb unloading induced a significant decrease in Akt, which is consistent with the research of other scholars. In addition, the downstream mTOR activity of Akt is closely related to the fiber type of gastrocnemius muscle [61]. Some scholars have proved that the transcription of Akt/mTOR is involved in the production of new muscle [62]. In addition, Akt is negatively regulated by MSTN [63]. In our study, it was found that the protein of Akt and mTOR was significantly down regulated after hindlimb unloading, and LIPUS with different intensities significantly promoted the expression of Akt, and 80 mW/cm^2^ LIPUS significantly promoted the expression of mTOR, indicating that MSTN/Akt/mTOR signal pathway may be the key molecular mechanisms of LIPUS preventing muscular atrophy caused by hindlimb suspension.

Kumar et al. [64] confirmed the up regulation of phenylalanine, tyrosine and tryptophan biosynthesis in the process of muscle cell differentiation, suggesting that these metabolites have important significance in muscle regeneration and pathology. In the muscle atrophy of hind limb unloading, we found that all the substances in this metabolic network decreased significantly. In our metabolic analysis, we found that L-tyrosine, as a key metabolite, would decrease significantly due to hind limb unloading, which would affect tryptophan biosynthesis. Tryptophan has been proved to be related to myogenic differentiation, and myogenic Akt1 and Akt2 will increase when tryptophan is supplemented in vitro, which creates favorable conditions for myogenic differentiation and myotube maturation [63,65,66,67].

In addition, alanine, aspartate and glutamate metabolism were also significantly affected by hind limb unloading. A significant decrease of alanine in hind limb unloading rats was detected. A cross-sectional study concluded that low level of alanine can be used to predict low muscle strength of the elderly, and the decrease of alanine amino transferase was detected in the blood. At the same time, the linear relationship between low muscle strength and low level of alanine amino transferase was verified [68]. Therefore, increasing the level of L-Alanine has a positive effect on promoting the growth of muscle strength. Both 30 mW/cm^2^ and 80 mW/cm^2^ LIPUS significantly increased the serum L-alanine level of hind limb unloading rats. In this study, we found that succinic acid as a key metabolite affects the metabolic balance, which is similar to the results of skeletal muscle metabolic abnormality model, in which succinic acid was found to induced the biosynthesis of mitochondria, promote the increase of the number of mitochondria, and enhance the respiratory capacity of oxidative skeletal muscle, and proposed that the significant decrease of succinic acid content in hind limb unloading may be related to the biosynthesis [69].Aspartic acid plays a variety of functions in physiological and biological processes and is a non-essential amino acid in mammals [70].Aspartic acid is used for the biosynthesis of nucleotides and proteins, and also plays an important role in skeletal muscle metabolism [71]. For example, aspartic acid supplementation can regulate muscle glucose uptake and improve muscle utilization of excess fatty acids and glycogen in rats [72,73]. However, whether it has an impact on muscle atrophy is still being explored. In our results, hind limb unloading can lead to a significant decrease in aspartate levels. LIPUS with different sound intensities can reduce muscle atrophy by promoting aspartate levels. The third pathway affected by hindlimb unloading was glycine, serine and threonine metabolism. In our results, the level of glycine was significantly decreased after hindlimb unloading. Similarly, the glycine level in the mouse model of muscular atrophy is decreased, and glycine supplementation can maintain the muscle mass and metabolic function of the mouse model of muscular atrophy [74,75]. Different from 30 mW/cm^2^, 80 mW/cm^2^ also significantly improved the level of metabolites in this pathway. Some scholars have come to the conclusion that when exogenous glycine is added to myoblast C2C12, the morphology of myotubes treated with glycine is larger, suggesting that increasing the utilization of glycine may help to reduce muscle consumption [76]. The effect of L-serine on muscle atrophy has not been studied, but some scholars found that serine treatment increased the expression of genes related to mitochondrial biogenesis, enhanced the quality and function of mitochondria, and improved the activity of myotubes of C2C12 cells [77]. Threonine is related to muscle protein turnover and overall metabolism [78,79]. The 80 mW/cm^2^ LIPUS can significantly promote the expression of the markers of this pathway, and the effect of 80 mW/cm^2^ LIPUS in improving gastrocnemius atrophy is slightly better, which may be related to its improvement of two metabolic pathways.

In conclusion, this study is the first time to demonstrate that LIPUS with different intensities can prevent muscle atrophy and promote myoblast proliferation by activating alanine, aspartate and glutamate metabolism and glycine, serine and threonine metabolism pathways in hindlimb unloading rats. In addition, LIPUS could inhibit the expression of MSTN and its receptor during hindlimb unloading. However, the exact relationship between LIPUS, MSTN and metabolic pathways still needs to be further verified. This study shows that LIPUS could be used to prevent muscle atrophy in the condition of microgravity or disuse for a long time, and can also be used as an adjuvant treatment for muscle atrophy.

## 4. Method

### 4.1. Animals

Forty-eight male Sprague Dawley rats (180–220 g) were obtained from the Laboratory Animal Breeding and Research Center of Xi’an Jiaotong University (Xi’an, China) and were housed in a controlled room (22 ± 2 °C, 60 ± 5% humidity, and 12-h light/dark cycle). All experiments were conducted with the approval of the Animal Ethical Committee of Shaanxi Normal University and in accordance with the Guide for the Care and Use of Laboratory Animals published by the US National Institutes of Health (NIH Publication No. 8023, revised in 1978).

### 4.2. Animal Modeling and Grouping

In this study, the model of muscle atrophy was induced by unloading the hind limbs of rats [80]. The tail of the rats was suspended, and the body was at an angle of 30° with the ground. The forelimbs could move freely along the bottom of the cage. During the experiment, they could eat and drink water freely. The rats were randomly divided into 4 groups: normal control group (NC, *n* = 8), hind limb unloading group (UL, *n* = 8), 30 mW/cm^2^ LIPUS irradiation group (UL + 30 mW/cm^2^, *n* = 8), 80 mW/cm^2^LIPUS irradiation group (UL + 80 mW/cm^2^, *n* = 8). The tails of rats in hindlimb unloading group were suspended for 28 days.

### 4.3. LIPUS Irradiation

The LIPUS equipment was self-developed and self-manufactured by the Shaanxi Key Laboratory of Ultrasonics (Shaanxi Normal University, Xi’an, China). After shaving, the right and left gastrocnemius of tail suspended rats were both irradiated with LIPUS for 28 d. The LIPUS parameters were 30 mW/cm^2^ or 80 mW/cm^2^, the center frequency was 1 MHz, the duty cycle was 20%, and the duration was 20 min/d.

### 4.4. Sample Preparation

The weight of rats was monitored every week. After 28 days of LIPUS irradiation, the rats were euthanized with an overdose of pentobarbital. Blood samples were obtained via abdominal aortapuncture, and then centrifuged at 1500× *g* for 20 min at 4 °C. The serum was separated and stored at −80°C until analysis. The gastrocnemius muscles were harvested and weighed. The left gastrocnemius muscle of each group was stored in 4% paraformaldehyde for morphological experiment. The right gastrocnemius muscle of each group was stored in liquid nitrogen for western blot and real-time quantitative PCR.

### 4.5. Fast and Slow Muscle Immunofluorescence

The muscle tissue was removed from 4% paraformaldehyde and dehydrated with sucrose solution. After frozen section, the sections were sealed with 3% BSA for 30 min, and then incubated with the first antibody (Servicebio, GB 112130/GB 111875, 1:3000/1:500) at 4 °C overnight and the fluorescent second antibody at room temperature for 50 min (Servicebio, gb21303/gb25301, 1:300/1:400). The nuclei were dyed with DAPI (Servicebio, g1012). The images were taken under an inverted fluorescence microscope (Olympus Bx-51, Tokyo, Japan) at ×200 magnification. Eight fields were randomly selected in each group, and the cross-sectional area of muscle fibers was calculated by Caseviewer 2.1. The average cross-sectional muscle fiber area was calculated as the total cross-sectional area/total number of muscle fibers in the visual field.

### 4.6. Muscle Tension In Vivo

The tension generated by the gastrocnemius muscle in vivo was measured by muscle test system instrument (1305A, Aurora, ON, Canada). Before the test, the rats were anesthetized in abdominal cavity, and supine on the rat platform in human anatomical posture, and the lower leg and thigh were 90 degrees, and the right knee was fixed with screws. The right sole of the rat’s right foot was fixed on the pedal with medical adhesive tape to ensure that the fixed position of the rat’s right foot would not be damaged by the contraction of the calf muscle during the current stimulation. Two metal electrodes were located at the nerves of gastrocnemius muscle to ensure that gastrocnemius could receive electric stimulation. The instrument runs the test program, and sends 30 Hz, 50 Hz, and 80 Hz electrical stimulation to gastrocnemius muscle respectively through the electrode, so as to make the gastrocnemius muscle to produce tension, and record the corresponding gastrocnemius muscle tension during electrical stimulation at each frequency, the unit is mN. Each rat repeated the test three times, and the values were statistically analyzed.

### 4.7. Cell Culture and Differentiation

Mouse myoblastic cell lineC2C12 cells were obtained from Xi’an Jiaotong University (Xi’an, China). The cells were cultured with Dulbecco’s Modified Eagle Medium (DMEM, HyClone) containing 10% fetal bovine serum (FBS, Gibco), 100 U/mL penicillinand 100 μg/mL streptomycin (1% P/S, sigma), at 37 °C in a humid atmosphere of 5% CO_2_. In order to induce myoblast fusion, when the cells grew to 80% confluence, the medium was changed to the differentiation medium of DMEM containing 2% heat inactivated horse serum (GIBCO) to induce differentiation. The medium was changed every other day for the next 5 days.

### 4.8. CCK-8 Detection

The myoblast activity was tested by CCK-8 kit (Dojindo Laboratories, Kumamoto, Japan). The cells were seeded onto 96-well plates. After 5 days of LIPUS treatment, the cells were incubated with CCK-8 working solution in the incubator for 2 h. The absorbance at 450 nm was detected by Model680 microplate reader (Belad, Philadelphia, PA, USA).

### 4.9. DAPI Staining

The cells were treated with LIPUS for 5 days. On day 6, the cells were fixed with 4% paraformaldehyde and stained with anti-fluorescence quenching blocking agent containing DAPI (Solarbio) for 10 min. Images capture was performed with an inverted fluorescence microscope (Olympus Bx-51, Tokyo, Japan) at ×400, and Image J was used to count the number of nuclei randomly selecting 8 fields for each group.

### 4.10. HE Staining

The cells were cultured in 24-well plates to induce differentiation. After changing 2%horse serum, the cells were irradiated with LIPUS for 5 days. On Day 6, after fixed with 95% ethanol for 20 min, the cells were successively stained with hematoxylin for 2 min and eosin staining for 1 min. Then dry the acquired image (200×). Eight different images were randomly collected from each well under an inverted optical microscope (Olympus Bx-51, Tokyo, Japan). The myotubes with two or more nuclei were counted using Image J.

### 4.11. Microfilament Immunofluorescence Staining

After treated with LIPUS for 5 days, the cells were fixed with 4% formaldehyde for 15 min, washed with PBS, and permeated with 0.2% Triton X-100 solution for 5min. After washed with PBS, the cells were incubated with FITC-phalloidin solution (Proteintech) to label microfilaments for 20 min at room temperature and washed with PBS. The nuclei were stained with DAPI for 10 min. Subsequently, photographs were taken under an inverted fluorescence microscope (Olympus Bx-51, Tokyo, Japan) (400×). The cell length and width were measured using image J, and finally the ratio of length/width was calculated. Three visual fields were measured in each group.

### 4.12. Western Blot

Weigh 0.05 g of gastrocnemius muscle tissue and homogenize it with 450 uL Ripa lysate. The protein concentration in gastrocnemius muscle was measured using BCA protein analysis kit (Thermo Scientific, Waltham, MA, USA). The equivalent total protein was electrophoretic in SDS- polyacrylamide gel (10–12%) and then transferred to nitrocellulose membrane. Immunoblotting was incubated with primary antibody at 4 °C overnight, and then incubated with corresponding secondary antibody (Cell signaling technology, Inc., Beverly, MA, USA) at room temperature for 1 h. Immunoreactive proteins were detected by enhanced chemiluminescence (ECL; Emerson). Azure biosystems C300 imaging system (Azure biosystems, CA, USA) was used to capture protein bands, and Image J software was used to quantify optical density. The following antibodies were used: MSTN (ab124721) and ActrIIb (ab180185) from Abcam, Akt (CST 9272s), mTOR (CST 2972s), and GAPDH (CST 14C10) from Cell Signaling Technology. The experimental method of cells was the same as above, MyHC3 primary antibody (Proteintech 22287) was used.

### 4.13. RNA Isolation and RT-PCR

The total RNA of gastrocnemius muscle was isolated by geneJET RNA Purification Kit (Thermo Scientific, Waltham, MA, USA). The cfx96 real time PCR system (Bio-Rad Laboratory, Hercules, CA, USA) was used to evaluate RNA integrity, and a 680 micro board reader (Bellard, Philadelphia, PA, USA) was used to quantify RNA integrity. Reverse transcription was carried out using the biological transcription Kit (Takara, Japan). The amplification procedure consisted of one cycle at 95 °C for 1 min, followed by 40 cycles at 94 °C for 15 s, 60 °C for 15 s, and 72 °C for 30 s. GAPDH was normalized as a housekeeping gene. The relative expression of gene was detected by RT-PCR. All experiments were repeated at least three times. The Primers are as follows: GATTATCACGCTACCACG (forward) and ATTCAGCCCATCTTCTCC (reverse) for MSTN, GCAGTCGTGGCAGAGTGAGCG (forward) and CTTGAGGTAATCCGTGAGGGAGC (reverse) for ActrIIB.

### 4.14. Metabonomicsan Analysis of Differential Metabolites and Pathways

A total of 500 μL serum was collected and stored in EP tube at −80 °C. During detection, 100 μL serum was taken, 300 μL methanol and 5 μL internal standard (9.9 mg/mL ribose alcohol) were added. Vortex mixing for 30 s and standing at −20 °C for 1 h. Centrifuge at 12,000 rpm for 10 min at 4 °C. 200 μL supernatant was collected and dried at room temperature for derivatization: 35 μL methoxypyridine solution (20 mg/mL) was subjected to strong vibration for 30 s, then oximation reaction was carried out at 37 °C for 90 min, and 35 μL BSTFA was added for derivatization, then reaction was carried out at 70 °C for 60 min, and then standing at room temperature for 30 min. GC/MS detection and analysis: GC/MS detection adopts multivariate statistical method to analyze the principal components. Meanwhile, through the orthogonal partial least squares discriminant analysis (OPLS-DA), the standard threshold for finding differential metabolites is VIP > 1, *p*-value < 0.05. All compounds are screened by Fiehn database for biomarkers, and the screened biomarkers are analyzed by MetPA database. Finally, the protein metabolic pathway was constructed.

### 4.15. Statistical Analysis

Results were expressed as mean ± SD. Statistical analysis was performed using SPSS version 20.0 (SPSS Institute, Chicago, IL, USA). Univariate analysis of variance among the four groups was performed and Tukey’s multiple comparison tests were used to determine the significance between each group. *p* values < 0.05 were considered statistically significant.

## Figures and Tables

**Figure 1 ijms-22-12112-f001:**
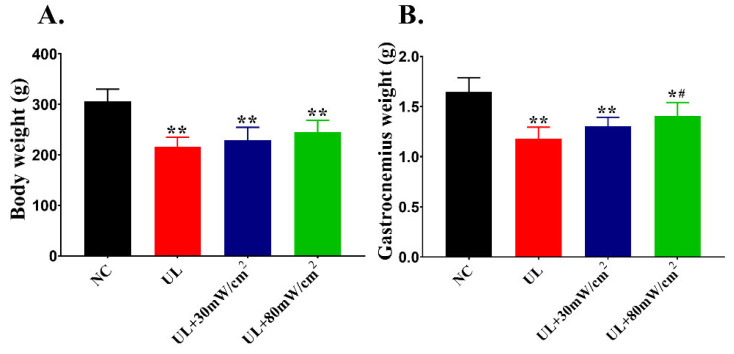
(**A**) Body weight. (**B**) Gastrocnemius muscle weight. The data were expressed as mean ± SD (*n* = 8 in each group), ** *p* < 0.01 vs. NC group, * *p* < 0.05 vs. NC group, ^#^ *p* < 0.05 vs. UL group.

**Figure 2 ijms-22-12112-f002:**
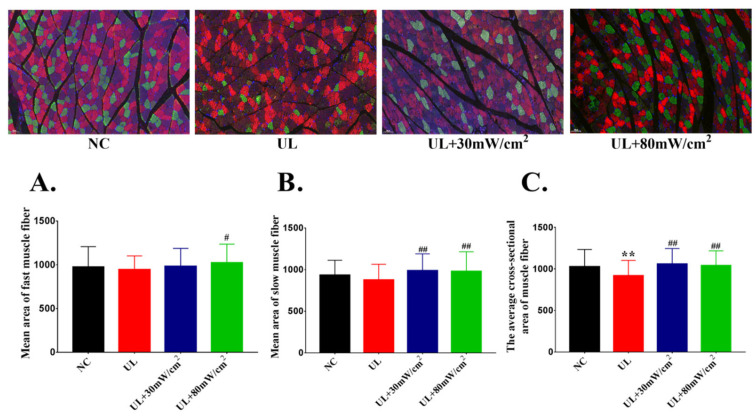
(**A**) The average cross-sectional area of fast muscle fibers. (**B**) The average cross-sectional area of slow muscle fibers. (**C**) The mean area of gastrocnemius muscle fiber. The data were expressed as mean ± SD (*n* = 8 in each group), ** *p* < 0.01 vs. NC group, ^##^ *p* < 0.01 vs. UL group, ^#^ *p* < 0.05 vs. UL group.

**Figure 3 ijms-22-12112-f003:**
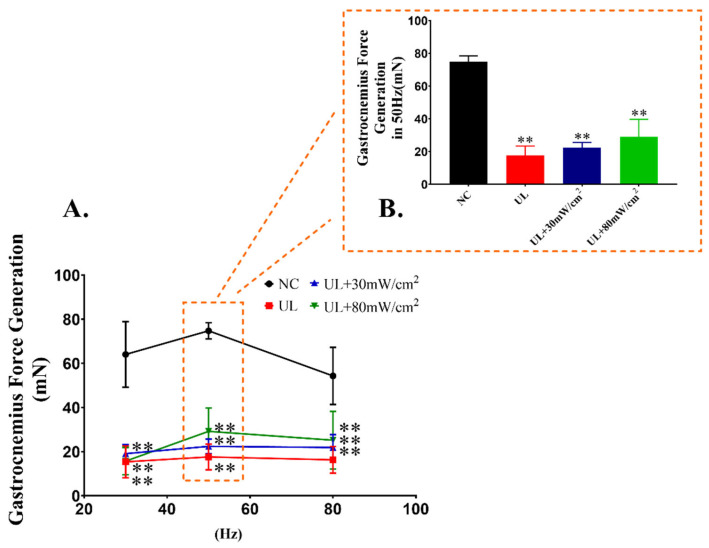
(**A**) The gastrocnemius force generation stimulated at 30, 50 and 80 Hz. (**B**) The gastrocnemius force generation stimulated at 50 Hz. The data were expressed as mean ± SD (*n* = 8 in each group), ** *p* < 0.01 vs. NC group.

**Figure 4 ijms-22-12112-f004:**
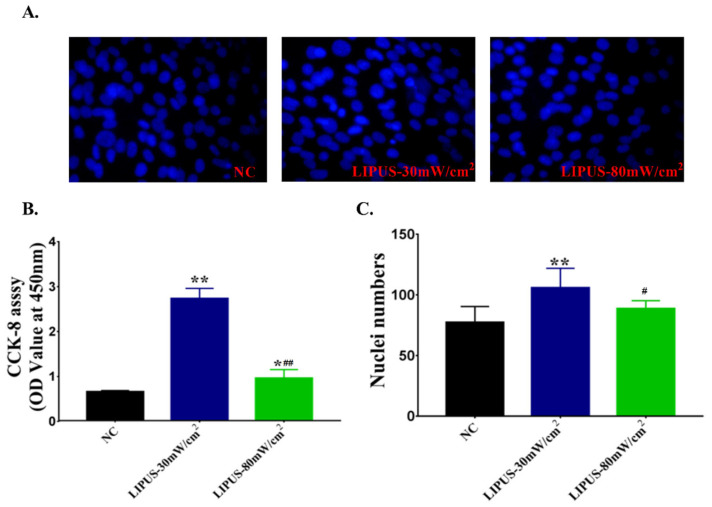
(**A**) DAPI staining of myoblasts C2C12 stimulated by LIPUS at different intensities. (**B**) The proliferation of C2C12 myoblasts. (**C**) The number of nuclei of C2C12 myoblasts. The data were expressed as mean ± SD (*n* = 8 in each group), ** *p* < 0.01 vs. NC group, * *p* < 0.05 vs. NC group, ^##^ *p* < 0.01 vs. LIPUS-30 mW/cm^2^group, ^#^ *p* < 0.05 vs. 30 mW/cm^2^ group.

**Figure 5 ijms-22-12112-f005:**
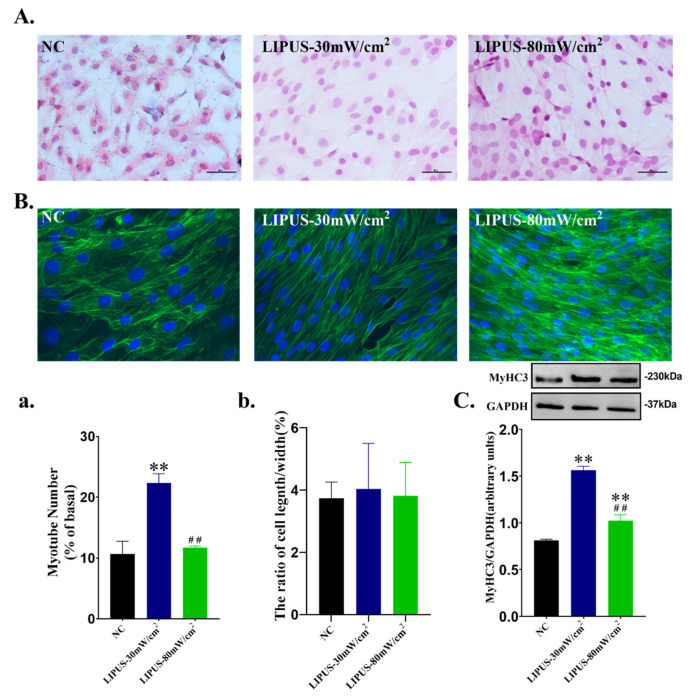
(**A**) HE staining of C2C12 after LIPUS irradiation with different intensities. (**B**) FITC-phalloidin labeled microfilament immunofluorescence staining. (**C**) MyHC3 protein expression of C2C12. (**A**(**a**)) The number of myotubes in HE staining (*n* = 8 in each group). (**B**(**b**)) The cell length/width ratio in microfilament staining (*n* = 3 in each group). ** *p* < 0.01 vs. NC group, ^##^
*p* < 0.01 vs. LIPUS-30 mW/cm^2^ group.

**Figure 6 ijms-22-12112-f006:**
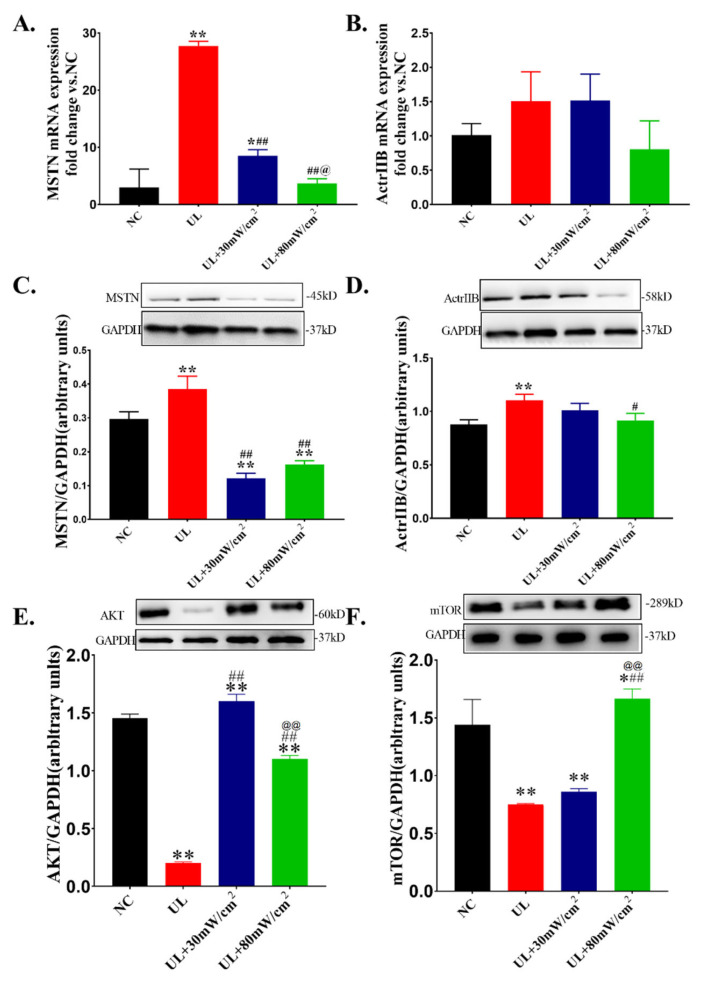
(**A**) The mRNA level of MSTN in gastrocnemius muscle. (**B**) The mRNA level of ActrΙΙB in gastrocnemius muscle. (**C**) The protein level of MSTN in gastrocnemius muscle. (**D**) The protein level of ActrΙΙB in gastrocnemius muscle. (**E**)The protein level of AKT in gastrocnemius muscle. (**F**) The protein level of mTOR in gastrocnemius muscle. The data were expressed as mean ± SD (*n* = 3 in each group), ** *p* < 0.01 vs. NC group, * *p* < 0.05 vs. NC group, ^##^ *p* < 0.01 vs. UL group, ^#^ *p* < 0.05 vs. UL group, ^@@^ *p* < 0.01 vs. 30 mW/cm^2^ group, ^@^ *p* < 0.05 vs. 30 mW/cm^2^ group.

**Figure 7 ijms-22-12112-f007:**
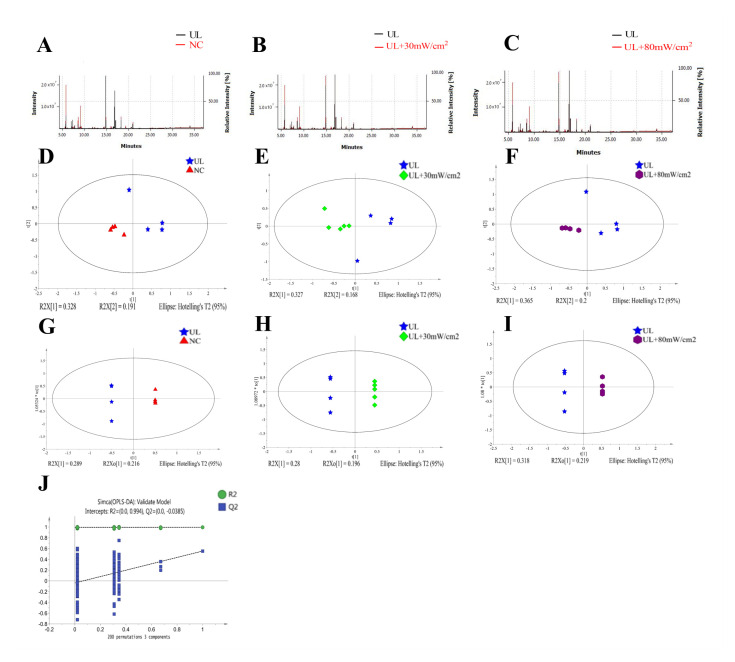
(**A**) Blood total ion flow diagram of NC and UL groups. (**B**) Blood total ion of UL and 30 mW/cm^2^ LIPUS groups. (**C**) Blood total ion flow diagram of UL and 80 mW/cm^2^ LIPUS groups. (**D**) PCA score of UL and NC group. (**E**) PCA score of UL and 30 mW/cm^2^ LIPUS groups. (**F**) PCA score of UL and 80 mW/cm^2^ LIPUS groups. (**G**) The OPLS-DA score of UL and NC rats. (**H**) The OPLS-DA score chart of UL and 30 mW/cm^2^ LIPUS groups. (**I**) The OPLS-DA score of UL and 80 mW/cm^2^ LIPUS groups was compared. (**J**) Validation of OPLS-DA model of rats blood samples from four groups by permutation test (the x-axis means the correlation coefficient between the original y variable and the permutated y variable and the y-axis is the value of R2 and Q2).

**Figure 8 ijms-22-12112-f008:**
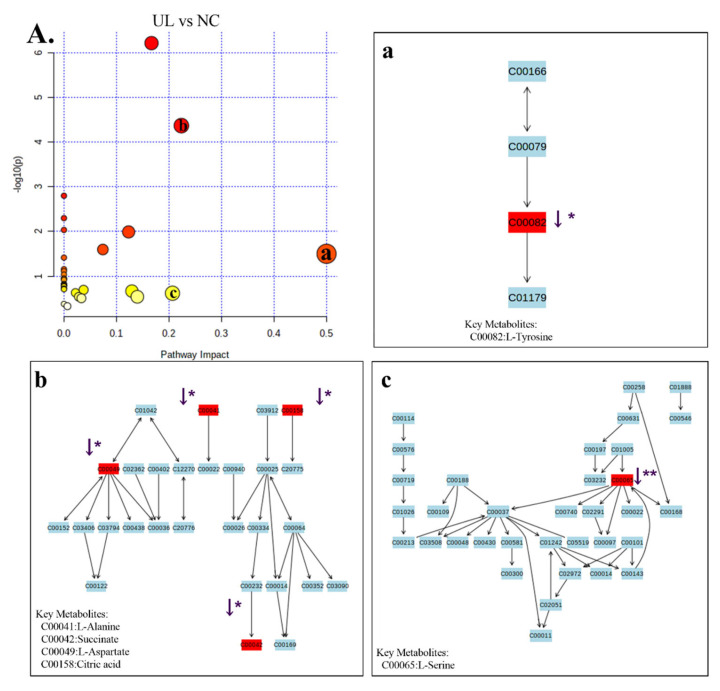
(**A**) Differential metabolic pathways between NC and UL groups. The color (light to dark) of the point represents the *p* value (Y-axis), and the size of the point represents the metabolic pathway impact (X-axis). (**A**(**a**)): phenylalanine, tyrosine and tryptophan biosynthesis; (**A**(**b**)): alanine, aspartate and glutamate metabolism; (**A**(**c**)): glycine, serine and threonine metabolism. * *p* < 0.05 vs. NC group, ** *p* < 0.01 vs. NC group.

**Figure 9 ijms-22-12112-f009:**
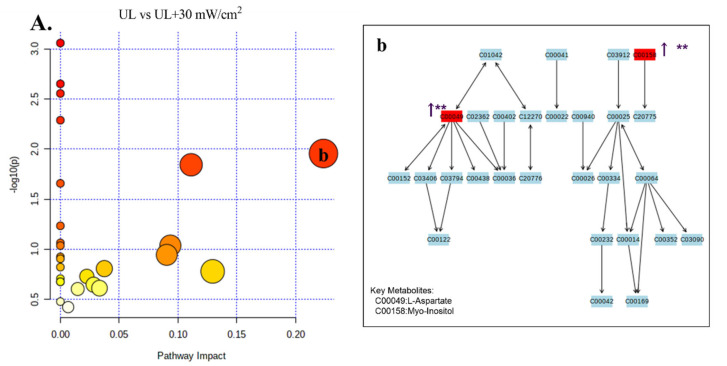
(**A**) Differential metabolic pathway between UL and 30mW/cm^2^ LIPUS groups. The color (light to dark) of the point represents the *p* value (Y-axis), and the size of the point represents the metabolic pathway impact (X-axis). (**A**(**b**)): alanine, aspartate and glutamate metabolism; ** *p* < 0.01 vs. UL group.

**Figure 10 ijms-22-12112-f010:**
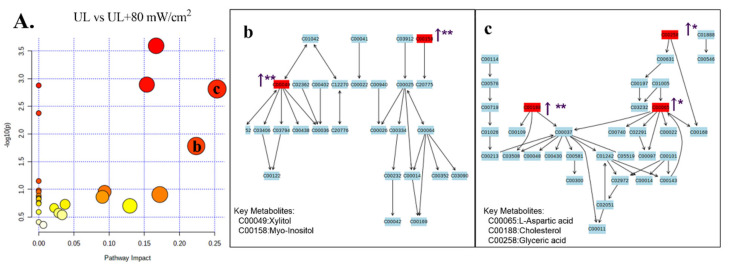
(**A**) Differential metabolic pathways between UL and 80 mW/cm^2^ groups. The color (light to dark) of the point represents the *p* value (Y-axis), and the size of the point represents the metabolic pathway impact (X-axis). (**A**(**b**)): alanine, aspartate and glutamate metabolism; (**A**(**c**)): glycine, serine and threonine metabolism. * *p* < 0.05 vs. UL group, ** *p* < 0.01 vs. UL group.

**Table 1 ijms-22-12112-t001:** Potential biomarkers and the relatedpathways.

Comparison	No.	RT	m/z	KEGG	Formula	Metabolite	Trend	Related Pathway
UL vs. NC	1	6.387	147.050	C00041	C3H7NO2	Alanine	↓ *	Alanine, aspartate and glutamate metabolism; ABC transporters
	2	9.490	149.050	C00042	C4H6O4	Succinic acid	↓ *	TCA cycle; oxidative phosphorylation; alanine, aspartate and glutamate metabolism
	3	10.985	130.050	C00049	C4H7NO4	L-Aspartic acid	↓ *	Alanine, aspartate and glutamate metabolism; serine and threonine metabolism; ABC transporters
	4	8.720	132.050	C00065	C3H7NO3	L-Serine	↓ **	Glycine, serine and threonine metabolism; sphingolipid metabolism
	5	16.436	179.050	C00082	C9H11NO3	Tyrosine	↓ *	Phenylalanine, tyrosine and tryptophan biosynthesis; phenylalanine metabolism
	6	8.495	189.050	C00086	CH4N2O	Urea	↓ **	Arginine and proline metabolism
	7	7.168	86.150	C00123	C6H13NO2	L-Leucine	↓ **	valine, leucine and isoleucine degradation; valine, leucine and isoleucine biosynthesis
	8	19.307	147.050	C00137	C6H12O6	Myo-Inositol	↓ *	Galactose metabolism
	9	15.779	73.050	C00158	C6H8O7	Citric acid	↓ *	TCA cycle
	10	6.112	72.150	C00183	C5H11NO2	L-Valine	↓ **	ABC transporters; Protein digestion and absorption
	11	5.748	106.050	C00186	C3H6O3	Lactic acid	↓ **	Gluconeogenesis; Pyruvate metabolism
	12	33.925	329.350	C00187	C27H46O	Cholesterol	↓ *	Steroid biosynthesis; Steroid hormone biosynthesis
UL vs. UL + 30 mW/cm^2^	1	10.985	130.050	C00049	C4H7NO4	L-Aspartic acid	↑ **	Alanine, aspartate and glutamate metabolism; arginine and proline metabolism
	2	8.509	73.050	C00086	CH4N2O	Urea	↑ *	Arginine and proline metabolism
	3	86.150	7.168	C00123	C6H13NO2	L-Leucine	↑ **	Valine, leucine and isoleucine degradation; valine, leucine and isoleucine biosynthesis
	4	226.150	19.338	C00137	C6H12O6	Myo-Inositol	↑ **	Galactose metabolism
	5	73.050	15.779	C00158	C6H8O7	Citric acid	↑ **	TCA cycle
	6	6.112	72.150	C00183	C5H11NO2	L-Valine	↑ **	ABC transporters; protein digestion and absorption
	7	33.920	129.050	C00187	C27H46O	Cholesterol	↑ *	Steroid biosynthesis; steroid hormone biosynthesis
	8	18.290	154.150	C00249	C16H32O2	Palmitic Acid	↑ ***	Fatty acid biosynthesis; fatty acid degradation
	9	9.837	89.050	C00258	C3H6O4	Glyceric acid	↓ *	Glycine, serine and threonine metabolism
	10	14.597	73.050	C00379	C5H12O5	Xylitol	↑ *	Pentose and glucuronate interconversions
UL vs. UL + 80 mW/cm^2^	1	14.597	73.050	C00049	C4H7NO4	Xylitol	↑ *	Alanine, aspartate and glutamate metabolism; arginine and proline metabolism
	2	10.985	130.05	C00065	C3H7NO3	L-Aspartic acid	↑ *	Glycine, serine and threonine metabolism; sphingolipid metabolism
	3	8.711	116.05	C00086	CH4N2O	L-Serine	↑ *	Arginine and proline metabolism
	4	8.506	137.05	C00123	C6H13NO2	Urea	↑ ***	Valine, leucine and isoleucine degradation; valine, leucine and isoleucine biosynthesis
	5	7.168	86.15	C00137	C6H12O6	L-Leucine	↑ *	Galactose metabolism
	6	19.307	217.15	C00158	C6H8O7	Myo-Inositol	↑ *	TCA cycle
	7	15.779	73.05	C00186	C3H6O3	Citric acid	↑ *	Gluconeogenesis; pyruvate metabolism
	8	5.742	6	C00187	C27H46O	Lactic acid	↑ *	Steroid biosynthesis; steroid hormone biosynthesis
	9	33.92	129.05	C00188	C4H9NO3	Cholesterol	↑ *	Valine, leucine and isoleucine biosynthesis; aminoacyl-tRNA biosynthesis
	10	9.245	117.05	C00249	C16H32O2	L-Threonine	↑ *	Fatty acid biosynthesis; fatty acid degradation
	11	9.807	133.050	C00258	C3H6O4	Glyceric acid	↑ *	Glycine, serine and threonine metabolism
	12	18.287	133.05	C00249	C16H32O2	Palmitic Acid	↑ **	Fatty acid metabolism
	13	6.116	55.15	C00183	C5H11NO2	L-Valine	↑ *	ABC transporters; Protein digestion and absorption

* *p* < 0.05 and ** *p* < 0.01, *** *p* < 0.001.

## Data Availability

The data that support the findings of this study are available from the corresponding author upon reasonable request.

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
