# Peer review of "Molecular and Metabolic Mechanism of Low-Intensity Pulsed Ultrasound Improving Muscle Atrophy in Hindlimb Unloading Rats"

_ijms, 2021, doi:10.3390/ijms222212112_

Round 1
Reviewer 1 Report
This manuscript reports the results of a study of the effect of low intensity pulsed ultrasound on rat muscle properties during hindlimb unloading. The authors probed the molecular mechanisms that contribute to ultrasound-induced reduction of muscle atrophy during simulated weightlessness using metabonomics approaches. The authors conclude that ultrasound can reduce muscle atrophy and propose that three identified metabolic pathways contribute to maintenance of muscle mass.
Comments and suggestions:
- The authors should state whether the right and left gastrocnemius muscles were both irradiated with LIPUS or if only one muscle in each rat was irradiated. If only one, was the other side studied as a control?
- End of line 17: “group” should be “groups”
- It is not exactly clear if the 28 days of ultrasound irradiation covered the entire duration of hindlimb suspension. The duration of suspension was not stated.
- Line 83: the Guide is published by the National Research Council, not the National Institutes of Health.
- Line 88: ”along the bottom of” should be inserted before “the cage”.
- Line 112: Change “nucleuses” to “nuclei”.
- Line 129: change “gastrocnemius muscle tension produce muscle tension” to “gastrocnemius muscle to produce tension”.
- Line 130: insert “during” before “electrical stimulation”.
- Line 143: The vendor(s) for the antibodies should be stated
- Lines 172-173: The sentence is incomplete and it is not clear. Were images capture with an inverted fluorescence microscope and were the numbers of cells in the images counted?
- Line 192: change “was used” to “were used”.
- The label on the Y-axis should be changed. “tesile tone” is poor terminology. Suggestion – “Gastrocnemius Force Generation”.
- Line 231: “The muscle tension of gastrocnemius muscle” should be changed to “The tension generated by the gastrocnemius muscle”.
- Line 232: change “rats” to “gastrocnemius”.
- The CCK-8 results are shown (Figure 4) but are not described in the Results section.
- The authors state, in the legend for Figure 4 that “LIPUS significantly promoted proliferation of C2C12 myoblasts”. That is a conclusion, not a result. The authors should simply state what Panel B shows – the number of nuclei.
- The legend for Figure 5 is essentially the same as what is stated in the text of the Results – lines 247-263. The authors should simply state in the legend what is shown in the figure, not the conclusions.
- The font size in Figure 6 is far too small.
- Figures 7, 8, and 9 – what do the different colors represent?
- Line 343: change “Long term lack of microgravity” to “Long term microgravity” or “Long term lack of gravity”.
- The authors state, multiple times throughout the manuscript, that LIPUS can improve muscle atrophy. This should be changed to reduces or prevents muscle atrophy.
- Line 365: the authors should clarify what they mean by slow fibers are used for energy metabolism or delete this.
- Line 375-377: The authors state that with a greater number of observations, there might be significant differences. This clearly indicates bias in their thinking and this should be deleted. It is equally possible that with a greater number of observations there will not be any significant differences.
- What is meant by “decrease of myocyte”?
Author Response
1.The authors should state whether the right and left gastrocnemius muscles were both irradiated with LIPUS or if only one muscle in each rat was irradiated. If only one, was the other side studied as a control?
Response: Thank you for your valuable comments. In this study, the right and left gastrocnemius muscles were both irradiated with LIPUS. The sentence of LIPUS irradiation in Method section was revised as “After shaving, the right and left gastrocnemius of tail suspended rats were both irradiated with LIPUS for 28 d.”. Besides, the related sentence in Abstract section was also revised as “The rats in the LIPUS treated group were simultaneously irradiated with LIPUS on gastrocnemius muscle in both lower legs at the sound intensity of 30 mW/cm2 or 80 mW/cm2 for 20 min/d for 28 days.”
2.End of line 17: “group” should be “groups”
Response: Thank you for your comments. It has been revised.
3.It is not exactly clear if the 28 days of ultrasound irradiation covered the entire duration of hindlimb suspension. The duration of suspension was not stated.
Response: Thank you for your comments. Yes, the 28 days of ultrasound irradiation covered the entire duration of hindlimb suspension. The duration of suspension is also 28 days. The sentence “The tails of rats in hindlimb unloading group were suspended for 28 days.” was added into Abstract and Method sections.
4.Line 83: the Guide is published by the National Research Council, not the National Institutes of Health.
Response: Thank you for your comments. After consulting some published papers (listed below) and the Guide for authors of several journals (Cytokine, Food and Chemical Toxicology, Oral oncology, Journal of the American Academy of Dermatology and so on), the Guide is published by National Institutes of Health.
Wang Z, Zheng Y, Wang F, et al. Mfsd2a and Spns2 are essential for sphingosine-1-phosphate transport in the formation and maintenance of the blood-brain barrier. Science Advances, 2020, 6(22):eaay8627.
Wang C, Zhu Y, Wu D, et al. The role of PDIA3 in myogenesis during muscle regeneration. Experimental & Molecular Medicine, 2020, 52: 105–117.
Garbuz D G , Astakhova L N , Zatsepina O G , et al. Functional Organization of hsp70 Cluster in Camel (Camelus dromedarius) and Other Mammals[J]. Plos One, 2011, 6.
5.Line 88: ”along the bottom of” should be inserted before “the cage”.
Response: It has been inserted.
6.Line 112: Change “nucleuses” to “nuclei”.
Response: It has been changed.
7.Line 129: change “gastrocnemius muscle tension produce muscle tension” to “gastrocnemius muscle to produce tension”.
Response: It has been changed.
8.Line 130: insert “during” before “electrical stimulation”.
Response: It has been inserted.
9.Line 143: The vendor(s) for the antibodies should be stated
Response: Thank you for your comments. The vendor(s) for the antibodies has been added. “The following antibodies were used: MSTN (ab124721) and ActrIIb (ab180185) from Abcam, Akt (CST 9272s) and mTOR (CST 2972s) from Cell Signaling Transduction.”
10.Lines 172-173: The sentence is incomplete and it is not clear. Were images capture with an inverted fluorescence microscope and were the numbers of cells in the images counted?
Response: Thank you for your comments. This sentence has been revised as “Images capture was performed with an inverted fluorescence microscope (Olympus Bx-51, Japan) at ×400, and Image J was used to count the number of nuclei randomly selecting 8 fields for each group.”
11.Line 192: change “was used” to “were used”.
Response: It has been changed.
12.The label on the Y-axis should be changed. “tesile tone” is poor terminology. Suggestion – “Gastrocnemius Force Generation”.
Response: It has been revised.
13.Line 231: “The muscle tension of gastrocnemius muscle” should be changed to “The tension generated by the gastrocnemius muscle”.
Response: It has been changed.
14.Line 232: change “rats” to “gastrocnemius”.
Response: It has been changed.
15.The CCK-8 results are shown (Figure 4) but are not described in the Results section.
Response: The CCK-8 results are described in Cell proliferation of the Results section “As shown in Figure 4B, compared with the NC group, 30 mW/cm2 and 80 mW/cm2 LIPUS significantly promoted cell proliferation (p< 0.01; p< 0.05), and the effect of 30 mW/cm2 is better than that of 80 mW/cm2 (p< 0.01).”
16.The authors state, in the legend for Figure 4 that “LIPUS significantly promoted proliferation of C2C12 myoblasts”. That is a conclusion, not a result. The authors should simply state what Panel B shows – the number of nuclei.
Response: The legend for Figure 4 has been revised as “Fig. 4 (A) DAPI staining of myoblasts C2C12 stimulated by LIPUS at different intensities. (B) The proliferation of C2C12 myoblasts. (C) The number of nuclei of C2C12 myoblasts. The data were expressed as mean ± SD (n = 8 in each group), **P<0.01 vs. NC group, *P<0.05 vs. NC group, ##P<0.01 vs. 30 mW/cm2 group, #P<0.05 vs. 30 mW/cm2 group.” Besides, all figure legends have been revised.
17.The legend for Figure 5 is essentially the same as what is stated in the text of the Results – lines 247-263. The authors should simply state in the legend what is shown in the figure, not the conclusions.
Response: The legend for Figure 5 has been revised as “Fig. 5 (A) The mRNA level of MSTN in gastrocnemius muscle. (B) The mRNA level of ActrⅡB in gastrocnemius muscle. (C) The protein level of MSTN in gastrocnemius muscle. (D) The protein level of ActrⅡB in gastrocnemius muscle. (E) The protein level of AKT in gastrocnemius muscle. (F) The protein level of mTOR in gastrocnemius muscle. The data were expressed as mean ± SD (n = 3 in each group), **P<0.01 vs. NC group,*P<0.05 vs. NC group, ##P<0.01 vs. UL group, #P<0.05 vs. UL group, @@ P<0.01 vs. 30 mW/cm2 group, @P<0.05 vs. 30 mW/cm2 group.” Besides, all figure legends have been revised.
18.The font size in Figure 6 is far too small.
Response: Figure 6 has been improved.
19.Figures 7, 8, and 9 – what do the different colors represent?
Response: The color (light to dark) of the point represents the p value (Y-axis), and the size of the point represents the metabolic pathway impact (X-axis). This sentence has been added into the legend in Figure 7, 8, and 9.
20.Line 343: change “Long term lack of microgravity” to “Long term microgravity” or “Long term lack of gravity”.
Response: It has been changed to “Long term lack of gravity”.
21.The authors state, multiple times throughout the manuscript, that LIPUS can improve muscle atrophy. This should be changed to reduces or prevents muscle atrophy.
Response: It has been revised according to your opinion.
22.Line 365: the authors should clarify what they mean by slow fibers are used for energy metabolism or delete this.
Response: This sentence has been deleted.
23.Line 375-377: The authors state that with a greater number of observations, there might be significant differences. This clearly indicates bias in their thinking and this should be deleted. It is equally possible that with a greater number of observations there will not be any significant differences.
Response: This sentence has been deleted according to your opinion.
24.What is meant by “decrease of myocyte”?
Response: Thank you for your comments. This sentence was not described clearly, which has been rewritten as “Tryptophan has been proved to be related to myogenic differentiation, and myogenic Akt1 and Akt2 will increase when tryptophan is supplemented in vitro, which creates favorable conditions for myogenic differentiation and myotube maturation”

Reviewer 2 Report
This paper shows some mechanisms of LIPUS improving muscle atrophy in hindlimb unloading rats. This study is potentially interesting. This paper will be strengthened by addressing the following issues.
In figure 4, the authors used myoblast C2C12. For skeletal muscle in vitro study, they need to include C2C12 myotube experiment data.
Author Response
In figure 4, the authors used myoblast C2C12. For skeletal muscle in vitro study, they need to include C2C12 myotube experiment data.
Response: Thank you for your valuable comments. The experiments about C2C12 myotube including HE staining, microfilament immunofluorescence staining and the protein expression of MyHC3 (Fig. 5) were added in the revised manuscript (as shown below). The methods of inducing myoblast fusion, HE staining, microfilament immunofluorescence staining and the protein expression of MyHC3 were added in the Method section. The results of the number of myotubes and the ratio of cell length/width and the protein expression of MyHC3 were described in the Results section. In the Discussion part, the related content and literatures were also added, so as the caption of Fig. 5.
